# Mapping and characterising electronic palliative care coordination systems and their intended impact: A national survey of end-of-life care commissioners

Jacqueline Birtwistle[1], Pablo Millares-Martin[2], Catherine J. Evans[3], Robbie Foy[1], Samuel Relton[1], Suzanne Richards[1], Katherine E. Sleeman[3], Maureen Twiddy[4], Michael I. Bennett[1], Matthew J. Allsop[1] *

1 Division of Primary Care, Palliative Care and Public Health, Leeds Institute of Health Sciences, University of Leeds, Leeds, United Kingdom, 2 Whitehall Surgery, Leeds, United Kingdom, 3 Cicely Saunders Institute of Palliative Care, Policy and Rehabilitation, King's College London, London, United Kingdom, 4 Hull York Medical School, University of Hull, Hull, United Kingdom

* m.j.allsop@leeds.ac.uk

**Data Availability Statement:** Pseudonymized data are stored on a secure server at the University of Leeds. Due to the sensitive information contained, restrictions to data have been mandated by the

## Abstract

### Objectives

In England, Electronic Palliative Care Coordination Systems (EPaCCS) were introduced in 2008 to support care coordination and delivery in accordance with patient preferences. Despite policy supporting their implementation, there has been a lack of rigorous evaluation of EPaCCS and it is not clear how they have been translated into practice. This study sought to examine the current national implementation of EPaCCS, including their intended impact on patient and service outcomes, and barriers and facilitators for implementation.

### Methods

We conducted a national cross-sectional online survey of end-of-life care commissioning leads for Clinical Commissioning Groups (CCGs) in England. We enquired about the current implementation status of EPaCCS, their role in information sharing and intended impact, and requested routine patient-level data relating to EPaCCS.

### Results

Out of 135 CCGs, 85 (63.0%) responded, with 57 (67.1%) having operational EPaCCS. Use of EPaCCS were confined to healthcare providers with most systems (67%) not supporting information sharing with care homes and social care providers. Most systems (68%) sought to facilitate goal concordant care, although there was inconsonance between intended impacts and monitoring measures used. Common challenges to implementation included healthcare professionals' limited engagement. Only one-third of patients had an EPaCCS record at death with limited recording of patient preferences.

National Health Service Health Research Authority and Health and Care Research Wales who oversaw ethical approval of our research. Requests for a de-identified dataset underpinning the findings presented in the manuscript can be made to the School of Medicine Ethics Committee at the University of Leeds, UK (contact can be made via email using FMHUniEthics@leeds.ac.uk).

**Funding:** This study is funded by the National Institute for Health Research (NIHR) [Health Services and Delivery Research (NIHR129171)]. The views expressed are those of the authors and not necessarily those of the NIHR or the Department of Health and Social Care. The funders had no role in study design, data collection and analysis, decision to publish, or preparation of the manuscript.

**Competing interests:** The authors have declared that no competing interests exist.

## Conclusions

Critical gaps exist in engagement with EPaCCS and their ability to facilitate information sharing across care providers. The limited alignment between stated goals of EPaCCS and their monitoring impedes efforts to understand which characteristics of systems can best support care delivery.

## Introduction

Timely coordination of care and treatment in the community is key to ensure individuals living with progressive chronic illness receive the right care, in the right place, at the right time [1]. Receipt of the right care can promote quality of life and enable individuals to remain in their preferred place of care, typically at home or in a care home [1–3]. In the United Kingdom (UK) Electronic Palliative Care Coordination Systems (EPaCCS) were introduced to support coordination and delivery of care in accordance with patient preferences. EPaCCS typically form part of an electronic clinical record system where information supporting delivery of a patient's care can be recorded and viewed. This record is generally initiated by healthcare professionals in the community and shared across healthcare settings to improve coordination of care for patients with progressive chronic illness, especially those nearing the end of life. Digital approaches to facilitate the collection, recording and sharing of information for palliative and end-of-life care are also being developed in the United States and Australia [4, 5]. This type of electronic sharing of patient clinical and administrative information across different systems and settings is referred to as Health information exchange (HIE). HIE is integral to visions to transform and modernise healthcare, to yield a more effective, efficient and personalised service [6].

EPaCCS have been seen as a key tool to enabling care coordination for palliative and end-of-life care in health policy for England since 2008 [7, 8]. In 2013, a survey by Public Health England indicated that up to 30% of all clinical commissioning groups (CCGs) had an operational EPaCCS, with 53% planning to implement systems [9]. However, significant regional variation was observed across the England regarding the content and delivery of EPaCCS across local clinical commissioning bodies [9]. Despite their implementation in care delivery, there is a limited evidence base underpinning their use. A recent systematic review highlighted that much of the evidence base on EPaCCS comprises expert opinion, and there is an absence of experimental studies evaluating the impact of EPaCCS on end-of-life outcomes [10]. Innovation to improve quality of care is desirable, but introducing a complex intervention at the interface of different stakeholders also requires an understanding of likely unintended consequences including patient harms [10]. Logical and well-intentioned policies and innovations can do more harm than good [11].

In most regions, EPaCCS comprise a template forming part of a patient's primary care electronic health record, with fields to capture preferences for care (e.g. do not resuscitate decision, and preferred places of care and death) where content is required to align with existing information standards (i.e. standards relating to the processing of information) [12]. However, little is known about the current extent of EPaCCS implementation, variations in their design, or access to systems across different care settings. A more detailed understanding of this complex intervention, operating at the interface of different healthcare providers and organisations, is required to identify common features that are perceived as having the potential to be provide additional benefit (or harm) to care delivery, and understand factors that enable or constrain

their implementation. Consequently, we conducted this research with the aim of examining how EPaCCS are being implemented, their intended impact, barriers and facilitators to their implementation, and processes for monitoring their uptake and use.

## Materials and methods

### Study design

We conducted a national cross-sectional exploratory survey. We report the study in accordance with the Checklist for Reporting Results of Internet E-Surveys (CHERRIES) [13] guideline for on-line survey distribution and reporting.

### Recruitment and sample

We surveyed all commissioning leads in palliative and end-of-life care in each Clinical Commissioning Group (CCG) in England (n = 135). Commissioning leads are typically either or both: i) a clinical lead, typically a GP, or; ii) a managerial lead in terms of the lead commissioning manager. Our exploratory survey approach sought responses from all CCGs, with an expected response rate of 53% (72/135 CCGs), based on a previous estimate from a meta-analysis of overall survey response rates among healthcare professionals [14]. A closed survey approach was adopted with a website link distributed via the clinical research networks in England to the respective commissioning lead for palliative and end-of-life care at each CCG. Where no palliative care lead could be identified, the research team made direct contact with the CCG. The survey was advertised through the National Health Service (NHS) England and the Improvement team for Palliative and End of Life Care team bulletin which is emailed directly to strategic, regional and clinical leads for palliative and end-of-life care across England. No incentives were offered.

### Questionnaire

Data were collected December 2020 –April 2021, using Online Surveys© (www.onlinesurveys.ac.uk, a secure survey platform developed by Jisc), and paper version when requested. The survey content was informed by an earlier systematic review by team members [10] and an earlier EPaCCS survey by Public Health England [9]. The survey was designed in consultation with palliative care and primary care clinicians within the project team alongside organisations with experience in developing standards for healthcare. Paper and on-line versions of the questionnaire that included up to 30 items that were piloted with senior health professionals working in palliative and primary care, such as clinical and commissioning leads for palliative and end-of-life care. Leads in each CCG were identified and sent a secure link to the online survey that was not detectable by search engines. Participants were first presented with information about the study and its aims and a consent form. Participants were required to provide written consent to participate prior to accessing the survey content. Following consent, respondents provided their name, email address and CCG enabling verification of responses. The online survey had adaptive questioning, designed to filter respondents to questions appropriate to whether their EPaCCS solution was operational, in the planning stage, or neither in place or planned for. Questions for CCGs planning EPaCCS implementation related to the planned system; CCGs without a system were asked about experienced and predicted challenges to implementation. Participants were able to review and change responses during completion. The questionnaire requested aggregated routine patient-level data from CCGs with an operational system for April 2019—September 2020, including numbers of deaths: i) in the CCG; ii) with an EPaCCS record; iii) with a record with preferred place of death recorded, and v) number with a record and a

diagnosis of cancer. Respondents could nominate another member of staff (e.g. an Information Manager) to provide this data at a later date. Items comprised closed-ended response format and open-ended items (see S1 File). Piloting suggested approximately 15 minutes was required to complete the survey. To maximize the response rate, participants were sent up to three reminders spaced one month apart. Responses submitted via the online survey platform were exported from the platform and stored securely on the systems of the lead institution.

### Analysis

Responses were checked for completeness and attributed to specific CCGs. We used descriptive statistics to derive frequencies and proportions of responses from each CCG and the remaining closed questions. Statistical analysis was conducted in SPSS. ArcGiS was used to generate a map plotting the registered address of each CCG to depict the status of EPaCCS (operational, in planning and no system) using colour-coding. The location of respondents and the status of their EPaCCS were plotted on a boundary and reference map (Clinical Commissioning Groups (April 2021) EN BUC) generated by the UK Office for National Statistics (Source: Office for National Statistics licensed under the Open Government Licence v.3.0; Contains OS data © Crown copyright and database right 2022. Microsoft Excel and Tableau version 2021.3 were used to produce box plots showing the median, interquartile range and upper and lower values for summary statistics relating to: i) patients with an EPaCCS record at death; ii) records with preferred place of death recorded, and; iii) proportion of EPaCCS records where a patient had a cancer diagnosis recorded. A directed content analysis approach [15] was undertaken (MA and JB) to analyse free text responses including quantification.

### Ethical approval

Ethical review was undertaken and approval granted by the North of Scotland Research Ethics Committee (research ethics committee reference, 21/NS/0046). All participants provided written informed consent after receiving an information letter detailing the wider project and its aims, what participation would entail and the handling, storage and use of data provided.

### Results

Eighty-two individuals responded on behalf of 85 of the 135 CCGs in England at the time of the survey (organisational response rate 63%) with no incomplete questionnaires, with some participants responding relating to more than one CCG. Responses for multiple CCGs by one respondent occurred in areas where delivery of palliative and end of life care is overseen across an Integrated Care System (a statutory partnership of organisations providing care across a geographical area that encompass more than one CCG). Across CCGs, EPaCCS were at different stages of implementation (Fig 1): 57/85 (67.1%) CCGs had operational EPaCCS, 15 (17.6%) had no EPaCCS and 13 (15.3%) were planning implementation. Operational EPaCCS had been introduced for differing durations including ≤ five years (n = 14/57; 24.6%) and ≥10 years prior (n = 7/57; 12.3%); 16/57 (28.0%) were unaware of the operational timeframe.

EPaCCS were hosted by different organisations across operational CCGs, including NHS trusts (n = 20/57; 35.1%), CCGs (n = 19/57; 33.3%), general practices (n = 7/57; 12.3%), and hospices (n = 5/57; 8.8%). A range of electronic health record system providers were involved in the storage and sharing of data captured in EPaCCS (see S1 Table). Across CCGs with operational EPaCCS, nine different electronic health record products were identified as being used to support the capture, storage or sharing of data. Across respondents, 86.0% (n = 49/57) hosted EPaCCS on more than one electronic health record system (median 3, IQR 2–5). The majority of systems (irrespective of software products used) did not facilitate patient access or

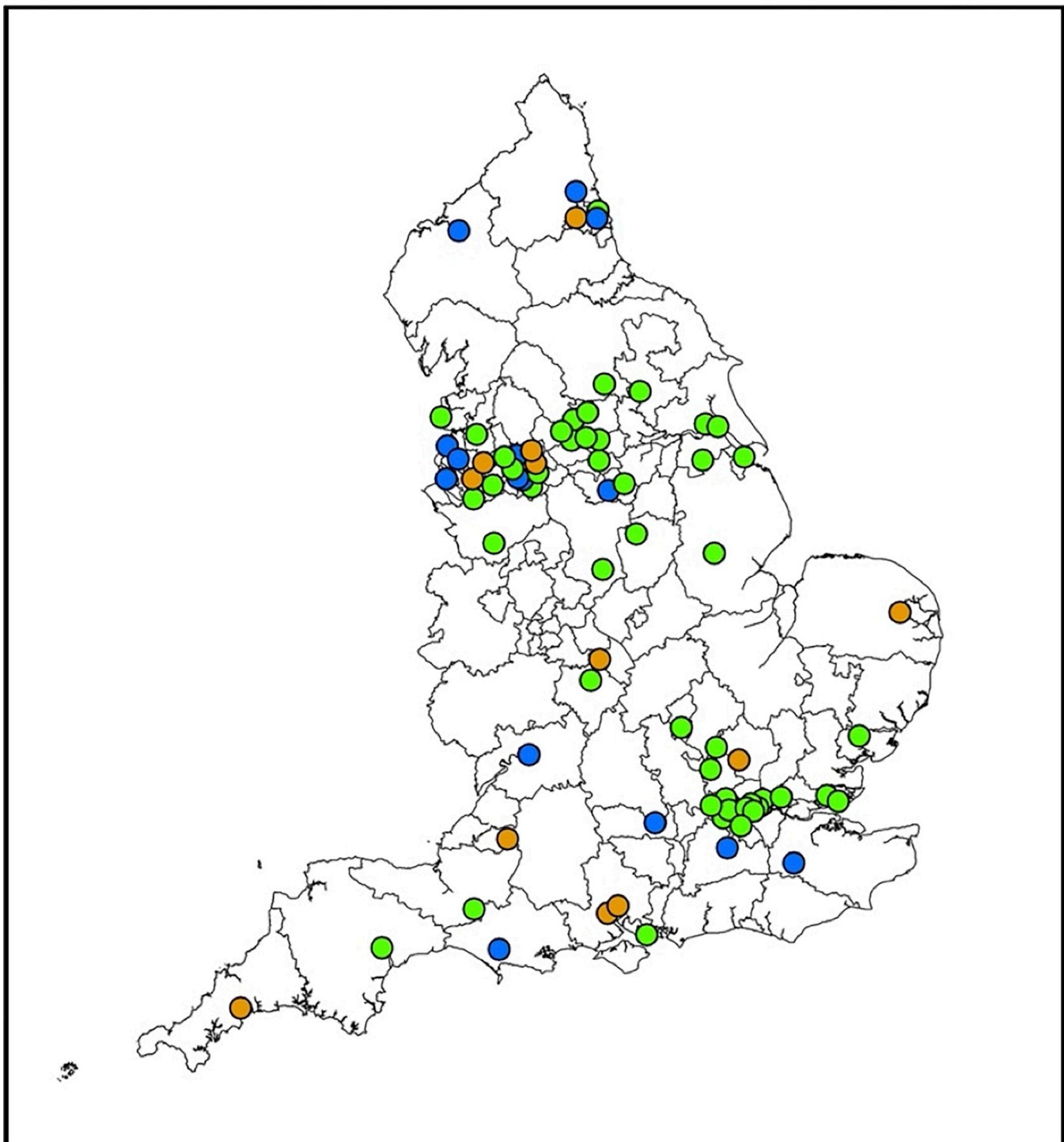

**Fig 1. Status of EPaCCS implementation across England. Key:** Colour-coding relates to implementation status of EPaCCS across CCGs using green (operational), orange (in planning) and blue (no system present).

editing of their own EPaCCS record with the exception of Coordinate My Care (CMC) a system used in London only. CMC enables patient access to EPaCCS records, although there was variation in the level of access reported by respective CCGs in London, ranging from patients

who could only view their record, to patients who could edit specific items (such as non-clinical information and care preferences) which would then require clinician review and approval.

## Sharing EPaCCS data across settings

Respondents reported access to EPaCCS across care settings (Fig 2, with further details of method of accessing EPaCCS outlined in S2 Table). Most CCGs (55/56 providing responses to this question; 98.2%) reported access by GPs and most community- and hospital-based palliative care teams. However, only 21% of CCGs reported care homes had EPaCCS access (i.e. 9/ 44 CCGs that detailed EPaCCS access in care homes).

## Intended impact and its measurement for operational or planned EPaCCS

Five key themes were derived from analysis of free-text responses regarding the intended impact of EPaCCS by CCGs (Table 1). The most commonly reported intended impact was to increase the likelihood of delivering care in accordance with patient wishes and priorities (68%), supporting continuity of care (49%) and improving timely access to documented and shared care plans (47%). Multiple methods currently used to measure intended impact were reported by CCGs with an operational EPaCCS (Table 1).

## Alignment of intended impact and its measurement

Fig 3 presents the alignment of intended impact ('high', 'low' or 'not measured') and CCG reported methods of measuring impact, with fewer than half of the measures cited had a high

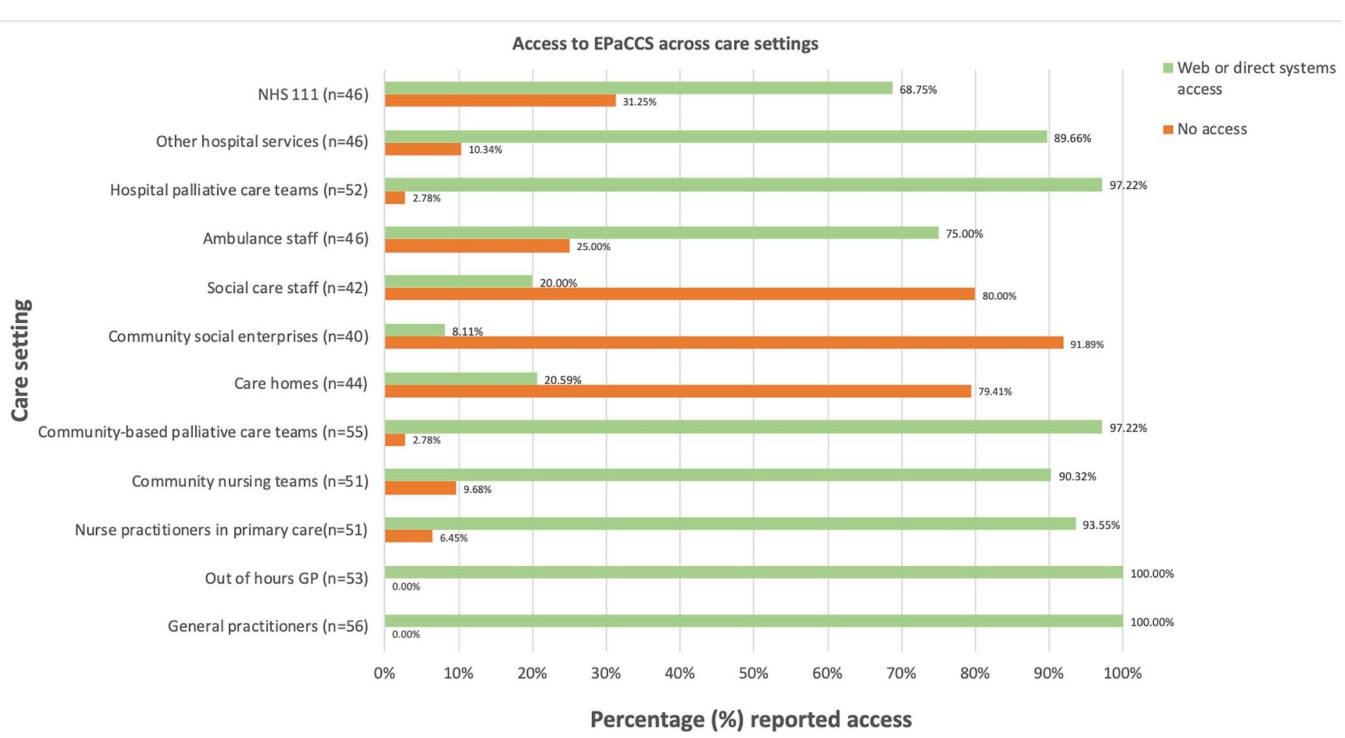

**Fig 2. Bar chart reflecting CCG reported access to EPaCCS by care setting. Key:** Percentage (%) reported access is derived from number of CCGs reporting access as a proportion of the number of CCGs providing a response. Of 57 respondents, 56 CCGs indicate a method of access for any of the services, with 23 CCGs providing data for some and not all health professional groups. In some CCGs, more than one method of sharing information was selected for the same service type. Data underpinning data in the graph can be found in S2 Table.

**Table 1. Intended impact of using EPaCCS and methods of measurement.**

| Theme | Intended Impact | Number of responses | | Total CCGs (N = 69) n (%) |
|---|---|---|---|---|
| | | With EPaCCS (N = 57) n (%) | Planning* (N = 12) n (%) | |
| **Access to information** | Timely access to documented and shared care plans and patient preferences for care | 28 (47) | 3 (25) | 31 (45) |
| **Care coordination** | Support coordination, continuity and delivery of patient-centred care between different health professionals and services. | 29 (49) | 9 (75) | 38 (55) |
| **Health professional practice** | Improve identification of patients with palliative diagnosis and in last year of life | 4 (7) | 0 | 4 (6) |
| **Family outcomes** | Improve experience of end-of-life care for families | 12 (20) | 1 (8) | 13 (19) |
| **Patient outcomes** | Increase likelihood of respecting patient wishes and priorities—*e.g. PPC/D, CPR* | 40 (68) | 9 (75) | 49 (71) |
| | Better conversations–(*e.g. appropriate timing and content*) | 22 (37) | 2 (17) | 24 (35) |
| **Types of data used to measure impact** | | | | |
| Concordance with patient stated preferences for place of care and death with attainment | | 29 (50) | 9 (75) | 38 (55) |
| Number of patients with an EPaCCS record | | 11 (19) | - | 11 (16) |
| Number of Hospital admissions and/or hospital attendances | | 10 (18) | 5 (42) | 15 (22) |
| Frequency of health professionals access to EPaCCS records | | 6 (11) | - | 6 (9) |
| Number of ambulance call-outs | | 5 (9) | - | 5 (7) |
| Completion of ACP information in EPaCCS records | | 5 (9) | 1 (8) | 6 (9) |
| Number of calls to community nurses or out of hours | | 1 (2) | - | 1 (1) |
| **Methods used to measure impact** | | | | |
| Feedback or surveys from health professionals and/or patients and families | | 20 (35) | 3 (25) | 23 (33) |
| Comparative analyses or benchmarking between CCGs (e.g. comparison of EPaCCS records across general practices in a CCG, dashboard linking impact on indicators to outcomes) | | 18 (32) | 3 (25) | 21 (30) |
| Audit (e.g. case note review of patients' EPaCCS data against the baselines and outcomes defined in local, regional and national standards, and retrospective death audits) | | 10 (18) | 2 (16) | 12 (17) |
| Case studies | | 10 (18) | - | 10 (14) |

* = No response from 1 CCG in planning. Counts represent the number of CCGs that mentioned each "impact" in the respective group for both those with EPaCCS and those with EPaCCS in planning. '-' indicates no data was provided for the category.

likelihood of accurately measuring the intended impact outlined by respondents. See S3 Table for detailed data on alignment between intended impact and measures in use reported by respondents.

## Barriers and facilitators to EPaCCS implementation

Responses (n = 18) from CCGs who had previously attempted implementation of a system indicated that problems with digital infrastructure and lack of health professional engagement had hindered implementation. Table 2 shows the reported challenges to the implementation or planned implementation of EPaCCS. Engagement of stakeholders with systems was the most commonly reported challenge.

Respondents (n = 52) described previous, current or planned activities that target clinical staff use and uptake of EPaCCS. Free-text responses (45 with operational EPaCCS, 7 with EPaCCS in planning) highlighted approaches to EPaCCS training including both one to one and virtual approaches. Training included development of comprehensive workbooks and bespoke training programmes for specific settings (e.g. care homes). Collaboration with teams in different services and settings within the CCG was important. Collaboration was often facilitated by end-of-life care networks with strong leadership and included engagement with

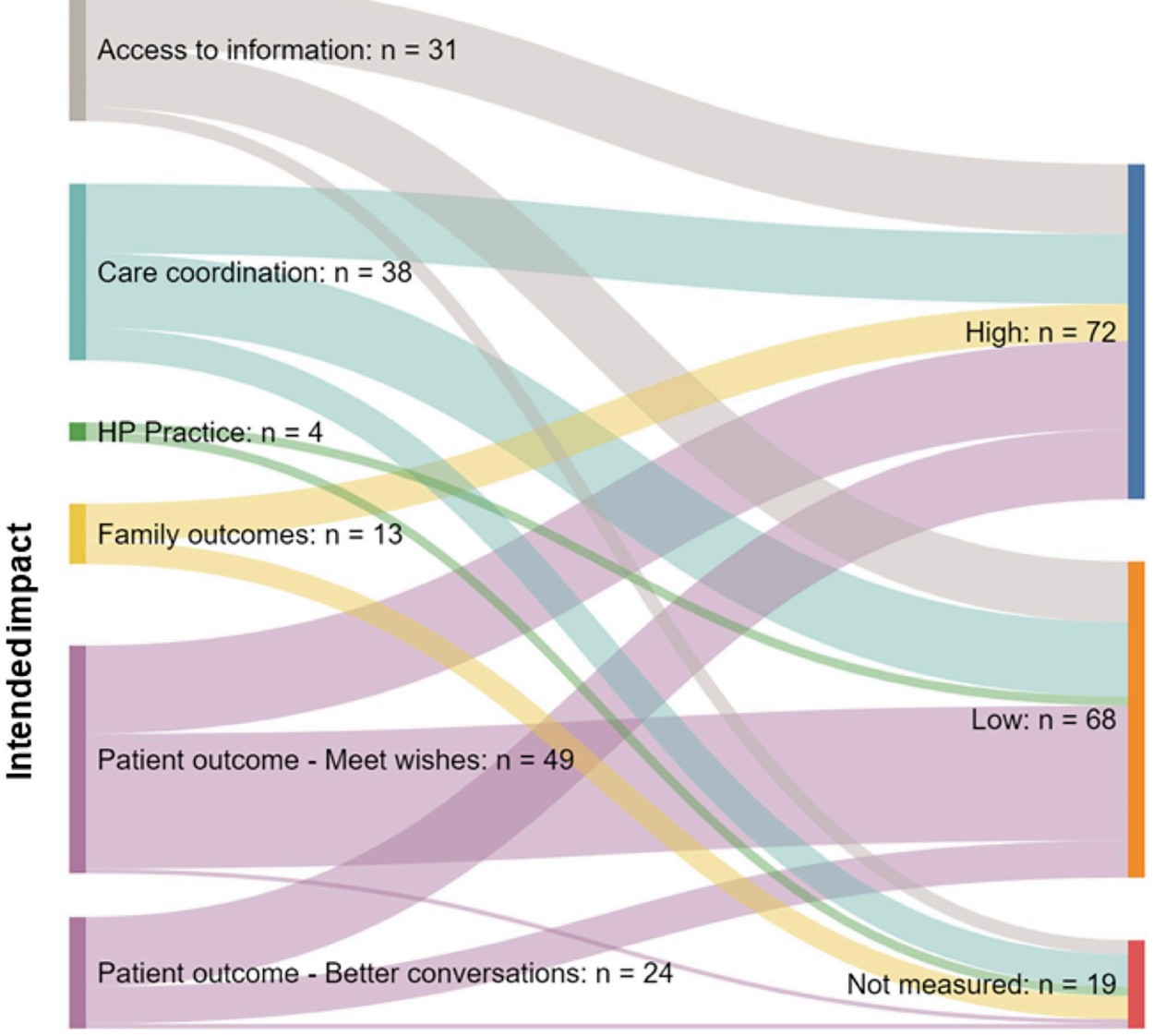

**Fig 3. Alignment of intended impact with measurement reported by survey respondents with an EPaCCS or with EPaCCS in planning. Key:** 'High' indicated that an appropriate range of measures were being used to measure the impact cited, including data on views, experiences or robust auditing methods linking individual patient outcomes to data. 'Low' indicated that methods had the potential to measure impact if being used robustly (e.g. on a patient by patient basis rather than aggregate routinely collected data). "Not measured' indicated that no measure was reported. Alignment between intended impact and measures used is outlined in detail in S3 Table.

other clinical services such as care homes, ambulance services and social care. Commitment at NHS Trust board level was also considered important, alongside assessing the acceptability of EPaCCS to increase uptake, engaging clinical facilitators or champions to drive staff use, having an active communication strategy, a dedicated project group or a consistent agenda item in palliative care team meetings.

### EPaCCS routine patient-level data

Routine data were provided for EPaCCS in 35/57 (61%) of CCGs, including data for 88,024 patient deaths from April 2019 –March 2020 (period 1) and 56,281 patient deaths in the period April 2020 –September 2020 (period 2). Fig 4 summarises the data reflecting variation in

**Table 2. Challenges to implementation.**

| | With EPaCCS n(%) | Planning n(%) | Total |
|---|---|---|---|
| Engagement with stakeholders (GPs) | 47 (82.5) | 8 (61.5) | 55 |
| Engagement with stakeholders (other) | 33 (57.9) | 4 (30.8) | 37 |
| Administration rights (i.e. issues with adding, administering or accessing records) | 22 (38.6) | 0 (0.0) | 22 |
| IT support | 18 (31.6) | 5 (38.5) | 23 |
| Training support | 16 (28.1) | 3 (23.1) | 19 |
| Patient consent | 14 (24.6) | 0 (0.0) | 14 |
| IT leadership | 14 (24.6) | 5 (38.5) | 19 |
| Clinical leadership | 12 (21.1) | 5 (38.5) | 17 |

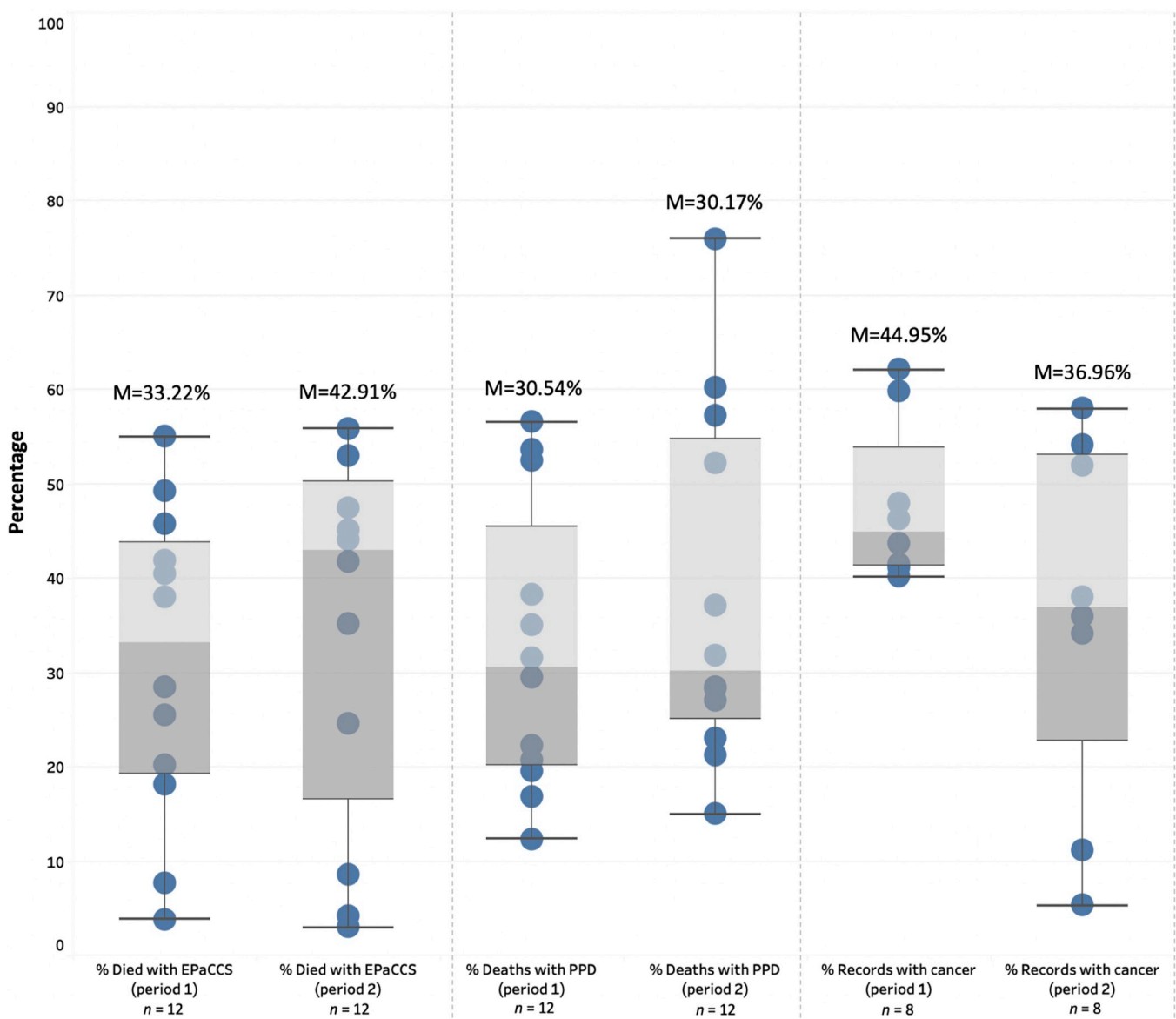

**Fig 4. Boxplot outlining range of patients with EPaCCS at death.** Key: M = median. PPD = preferred place of death. Percentage for category using data provided for reported using available data only.

practice across CCGs. For periods 1 and 2, the median proportion of people with an EPaCCS at death was 33.22% and 42.91%, respectively. In both periods, three CCGs reported <20% of patients having an EPaCCS at death. Of those people who died with an EPaCCS in place, the median proportion of a preferred place of death being recorded was 30.54% (period 1) and 30.17% (period 2). The proportion of patients with an EPaCCS record who had a cancer diagnosis in their record was a median of 44.95% in period 1 and 36.96% in period 2.

## Discussion

Despite policy supporting adoption of EPaCCS in England since 2008, a third of palliative and end-of-life care lead respondents indicated they were planning implementation in the future or had no EPaCCS. Where implemented, respondents outlined widespread variation in the extent of, and approaches to, EPaCCS implementation. Critical gaps in information sharing were identified, with most EPaCCS not facilitating access for key providers of palliative care, including care homes and social care staff. The intended impacts of operational EPaCCS are aligned with national policy goals of enhancing coordination of care, early identification and recording of people approaching end of life, and reducing avoidable hospital admissions [16]. However, over half of CCGs reported either undertaking no monitoring of impact, or using measures with limited alignment to the intended system impacts. Common challenges experienced with implementation of EPaCCS included difficulties with engaging health professionals involved in the entry and review of data in EPaCCS records and issues with access to and sharing of information, with systems often composed of multiple electronic clinical systems. From routine data provided by CCGs we determined that one third of patients had an EPaCCS record in place at death. Of those with an EPaCCS record, fewer than one third of patients had key information relating to their preferences for care recorded (i.e. a preferred place of death) and nearly half had a diagnosis of cancer. Key findings aligned with what is known and the implications of this work are summarised in Table 3.

**Table 3. Summary of key findings and their implications.**

| | |
|---|---|
| **What is already known on this topic?** | • Digital approaches to facilitate the collection, recording and sharing of information to support palliative and end-of-life care delivery are being developed in countries including the UK, United States and Australia<br>• Policy has supported the use of Electronic Palliative Care Coordination Systems (EPaCCS) in England since 2008<br>• It is not known which characteristics of EPaCCS are perceived as beneficial to care delivery or constraining to implementation |
| **What this study adds** | • There is considerable variation in how EPaCCS have been implemented across England<br>• Most EPaCCS do not allow sharing of information with care homes and social care staff, who often have central roles in end-of-life care<br>• There is limited alignment between the intended impact of EPaCCS and the current methods being used to monitor and assess whether impact is being realised<br>• Around one-third of people have an EPaCCS record at death and these are more commonly created for people with a diagnosis of cancer |
| **How this study might affect research, practice or policy** | • The proportion of people dying with an EPaCCS record does not meet conservative population-based estimates of palliative need, despite policy ambitions for EPaCCS to support early identification of patients<br>• Patient and caregiver perspectives on EPaCCS are lacking and should be incorporated into the development of systems to ensure they facilitate and support patient-centred care<br>• Future successful implementation and evolution of EPaCCS is likely to need stronger stakeholder engagement and better interoperability |

The use of templates to structure data collection and recording around specific diseases, such as those forming part of EPaCCS, are common across England and other healthcare systems [17]. Templates can improve documentation where used [18]. However, our findings indicate that, for those CCGs providing data, the majority of people are dying without an EPaCCS record. Population-based estimates of those who die that may benefit from palliative care range from 69%–82% [19]. Our study estimated a median of 33.22% and 42.91% of people who died had an EPaCCS record over two time periods. Even CCGs with the highest proportion of people dying with an EPaCCS record do not meet conservative population estimates of palliative need. This contradicts one of the untested assertions of EPaCCS being a tool to facilitate early identification and recording of people approaching end of life [16]. Low uptake may, in part, be linked to implementation challenges, including engagement of stakeholders (i.e. those responsible for updating and accessing EPaCCS records) and interoperability issues around accessing EPaCCS across care settings. Furthermore, EPaCCS need to be considered within the wider framework of advance care planning [20]. Multiple, requisite steps for advance care planning begin with patients being able to articulate, and clinicians elicit, values and preferences for care [21]. Completion of EPaCCS records is also susceptible to the known multiple barriers that impede advance care planning (e.g. need for sufficient time to have conversations and education needs around professional and legal responsibilities) [22]. Once created, an EPaCCS record is commonly hosted across multiple electronic clinical record software products, creating a complex informatics landscape within which data are stored and shared. Interoperability is an enduring challenge and remains a priority for health and social care delivery in the UK [18]. For EPaCCS, improvements in interoperability are required to ensure information sharing occurs across all providers involved in the delivery of specialist and core level palliative care in England that includes care homes and social care [23]. Gaps across other settings persist too, such as urgent care providers, despite evidence demonstrating a demand for EPaCCS in these settings [24].

There is considerable variation in the intended impact of EPaCCS across CCGs alongside limited alignment of measures informing whether intended impacts are being realised. Assessment of whether EPaCCS are supporting delivery of care in accordance with patient preferences will need to reflect specific contexts and intended impacts so a future gold standard is unlikely to be appropriate for all EPaCCS [25]. CCGs will require a range of indicators to capture the process of implementation and clinical outcomes. Including indicators for process benefits would align with evidence that these can be realised with templates embedded in electronic health records [26]. Clinical indicators require further research to determine causal pathways relating to how EPaCCS and the information it contains influences care delivery. CCGs with operational EPaCCS or systems in planning should consider collecting indicators that can inform the success (or otherwise) of its implementation that are sensitive to local context and population needs. This would address prior deficits identified across local authority policymaking for palliative care, where there has been a lack of alignment between specific populations, their needs, and interventions used [27]. Furthermore, the development of measures for monitoring EPaCCS and similar technology-mediated approaches to care coordination should consider equity. Our study identified that a limited number of EPaCCS records are being created, with limited information recorded. Monitoring how many people are receiving EPaCCS records and having care preferences recorded, shared and reviewed (and by whom) would provide important baseline data, including determining how this varies by ethnicity, deprivation and disease groups.

Most EPaCCS do not support patient access to their own records, with the exception of a system in London. However, increasing policy support for integrated electronic personal health records that enable access for patients and carers could affect EPaCCS [18]. A critical

gap in the evidence base to guide personal health record development is the absence of patient and caregiver preferences for information included within EPaCCS, and the degree to which they can access, review and edit their own records. Patient perspectives on approaches to health information exchange have been sought for mental health services which indicated patients vary in their willingness to trust others with personal health information, but may be willing to participate in such approaches because of perceived individual and societal benefits [28]. Understanding patient and caregiver perspectives on EPaCCS will be essential to ensure systems record and share information that can facilitate and support patient-centred care. While disease templates may improve documentation of key measures, there is also scope for them to restrict clinical review processes, risk health professionals' agendas being prioritised over those of patients [29], promote "bureaucratisation of care" and disregard aspects of quality care not considered within a template [30]. There may be scope to optimise EPaCCS implementation, but not without addressing the current dearth of patient and caregiver engagement.

This national survey of end-of-life care commissioning leads across CCGs in England exceeded our target response rate despite conducting the survey during the COVID-19 pandemic. Notwithstanding this, we acknowledge a potential response bias, with those with more established and operating EPaCCS more likely to respond, although reports from CCGs with no EPaCCS or systems in planning were gathered. Additionally, we assume respondents accurately recalled information relating to EPaCCS. Recruitment was affected by refusals to participate given demands on capacity relating to the pandemic and difficulties in identifying a named end-of-life care lead. Provision of routine clinical data was limited to 60% of respondents, with data access issues arising for some respondents resulting from planned reorganisation and merging of CCGs into integrated care systems. Routine data itself related to the pre-pandemic (period 1) and pandemic (period 2), with the latter timeframe including first and second waves of COVID-19 in England.

## Conclusions

There is considerable variation in how EPaCCS have been implemented across England and there remain challenges around stakeholder and end user engagement. Where EPaCCS are present, only a limited proportion of those who may be eligible for an EPaCCS record are receiving one before death, with limited recording of preferences for care. This may in part be influenced by widespread interoperability challenges and the lack of information sharing across settings (including care homes and ambulance trusts) that are integral components of palliative care and end-of-life care delivery. Despite policy advocating their use since 2008, the impact of EPaCCS remains largely unknown, their implementation challenging, and their uptake and use limited. As technology-mediated approaches to advance care planning continue to be developed internationally, future research is essential to understand if and how they can be implemented optimally in the delivery of palliative and end-of-life care.

## Supporting information

**S1 Checklist. Checklist for Reporting Results of Internet E-Surveys (CHERRIES).**
(DOCX)

**S1 File. Copy of the survey items sent to commissioners of palliative care and end of life services in England.**
(PDF)

**S1 Table. Electronic health record system providers involved in storage and sharing of data captured in EPaCCS.**
(PDF)

**S2 Table. Care settings and levels of access to EPaCCS as reported by CCGs with an operational system.**
(PDF)

**S3 Table. Alignment of intended impact with measurement reported by survey respondents CCGs with operational EPaCCS.**
(PDF)

## Acknowledgments

We thank all end-of-life care commissioners for their time and contribution to the research. We also thank Professor Julia Riley for coordinating access to data across the London region. We would like to acknowledge our project public and patient involvement group whose views and experiences helped to inform the survey questions. The survey research questions were also reviewed and discussed through consultation with local groups of palliative care patients, who reported no changes to survey items were required. Our project co-applicant and lead for patient and public involvement, Dr Barbara Hibbert, died during the development of this manuscript. Barbara supported the development of project plans, formed our patient involvement group, and facilitated inclusive and insightful patient and public involvement group meetings. She will be greatly missed by the team.

## Author Contributions

**Conceptualization:** Katherine E. Sleeman, Michael I. Bennett, Matthew J. Allsop.

**Data curation:** Jacqueline Birtwistle, Matthew J. Allsop.

**Formal analysis:** Jacqueline Birtwistle, Pablo Millares-Martin, Samuel Relton, Matthew J. Allsop.

**Funding acquisition:** Pablo Millares-Martin, Catherine J. Evans, Robbie Foy, Samuel Relton, Suzanne Richards, Katherine E. Sleeman, Maureen Twiddy, Michael I. Bennett, Matthew J. Allsop.

**Investigation:** Jacqueline Birtwistle, Pablo Millares-Martin, Michael I. Bennett, Matthew J. Allsop.

**Methodology:** Jacqueline Birtwistle, Pablo Millares-Martin, Catherine J. Evans, Robbie Foy, Samuel Relton, Suzanne Richards, Katherine E. Sleeman, Maureen Twiddy, Michael I. Bennett, Matthew J. Allsop.

**Project administration:** Jacqueline Birtwistle, Matthew J. Allsop.

**Supervision:** Michael I. Bennett, Matthew J. Allsop.

**Validation:** Jacqueline Birtwistle, Pablo Millares-Martin, Samuel Relton, Matthew J. Allsop.

**Visualization:** Jacqueline Birtwistle, Pablo Millares-Martin, Matthew J. Allsop.

**Writing – original draft:** Jacqueline Birtwistle, Pablo Millares-Martin, Matthew J. Allsop.

**Writing – review & editing:** Jacqueline Birtwistle, Pablo Millares-Martin, Catherine J. Evans, Robbie Foy, Samuel Relton, Suzanne Richards, Katherine E. Sleeman, Maureen Twiddy, Michael I. Bennett, Matthew J. Allsop.

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
