## [Decision Letter · Decision Letter 0]

20 Sep 2022

PONE-D-22-16285Mapping and characterising electronic palliative care coordination systems and their intended impact: A national survey of end-of-life care commissionersPLOS ONE

Dear Dr. Allsop,

Thank you for submitting your manuscript to PLOS ONE. After careful consideration, we feel that it has merit but does not fully meet PLOS ONE’s publication criteria as it currently stands. Therefore, we invite you to submit a revised version of the manuscript that addresses the points raised during the review process.

We look forward to receiving your revised manuscript.

Kind regards,

Mona Dür, PhD, MSc

Academic Editor

PLOS ONE

Journal Requirements:

This study is funded by the National Institute for Health Research (NIHR) [Health Services and Delivery Research (NIHR129171)]. The views expressed are those of the authors and not necessarily those of the NIHR or the Department of Health and Social Care.

6. We note that Figure 1 in your submission contain [map/satellite] images which may be copyrighted. All PLOS content is published under the Creative Commons Attribution License (CC BY 4.0), which means that the manuscript, images, and Supporting Information files will be freely available online, and any third party is permitted to access, download, copy, distribute, and use these materials in any way, even commercially, with proper attribution. For these reasons, we cannot publish previously copyrighted maps or satellite images created using proprietary data, such as Google software (Google Maps, Street View, and Earth). For more information, see our copyright guidelines: http://journals.plos.org/plosone/s/licenses-and-copyright.

Additional Editor Comments:

Your manuscript entitled "Mapping and characterizing electronic palliative care coordination systems and their intended impact: A national survey of end-of-life care commissioners" which you submitted to Plos One, has been reviewed. The reviewer comments are included at the bottom of this letter.

The reviews are in general favourable and suggest that, subject to minor revisions, your paper could be suitable for publication. Please consider these suggestions, and I look forward to receiving your revision.

Reviewers' comments:

Reviewer's Responses to Questions

**Comments to the Author**

1. Is the manuscript technically sound, and do the data support the conclusions?

Reviewer #1: Yes

Reviewer #2: Yes

2. Has the statistical analysis been performed appropriately and rigorously? 

Reviewer #1: Yes

Reviewer #2: I Don't Know

3. Have the authors made all data underlying the findings in their manuscript fully available?

Reviewer #1: Yes

Reviewer #2: No

4. Is the manuscript presented in an intelligible fashion and written in standard English?

Reviewer #1: Yes

Reviewer #2: Yes

5. Review Comments to the Author

Reviewer #1: Thank you for the opportunity to review this manuscript that examines an important policy translation issue in palliative care in UK. The paper is a well thought-out and well-written one. I have very minor comments.

Results highlighted that 82 individuals responded on behalf of 85 CCGs. Does this infer that some participants reported on behalf of more than one CCG? I think this needs to be clearly stated and the implications explored.

Line 13 on Page 7 reads: 'For operational CCGs, nine different electronic patient record software products were reported as used to host EPaCCS.'

This sentence needs to be rewritten for clarity.

Reviewer #2: Thank you for giving me opportunity for reviewing this paper. It is an important article about use and impact of electronic palliative care co-ordination system. However, I have some recommendation regarding the manuscript.

1. In the abstract, it would be better if you mention from whom or where you have collected the data (eg. health care providers/ patients etc) in the methods section.

2.Same comment goes for the recruitment and sample section of the methodology

3. Have you conducted any statistical testes in the tables no 1 and 2. Some statistical tests can make your results more stronger

4. I think table 3 in the discussion section can be moved to the appendix section. Also the points inside the tables can be described under the heading of strength and limitations

6. PLOS authors have the option to publish the peer review history of their article (what does this mean?). If published, this will include your full peer review and any attached files.

Reviewer #1: No

Reviewer #2: No

---

## [Author Response · Author response to Decision Letter 0]

23 Sep 2022

Please see attached our point-by-point response to reviewer and editor comments.

---

## [Editor Report · Decision Letter 1]

27 Sep 2022

Mapping and characterising electronic palliative care coordination systems and their intended impact: A national survey of end-of-life care commissioners

PONE-D-22-16285R1

Dear Dr. Allsop,

We’re pleased to inform you that your manuscript has been judged scientifically suitable for publication and will be formally accepted for publication once it meets all outstanding technical requirements.

Kind regards,

Mona Dür, PhD, MSc

Academic Editor

PLOS ONE
---

## [Editor Report · Acceptance letter]

5 Oct 2022

PONE-D-22-16285R1 

Mapping and characterising electronic palliative care coordination systems and their intended impact: A national survey of end-of-life care commissioners 

Dear Dr. Allsop:

I'm pleased to inform you that your manuscript has been deemed suitable for publication in PLOS ONE. Congratulations! Your manuscript is now with our production department. 

Kind regards, 

on behalf of

Dr. Mona Dür 

Academic Editor

PLOS ONE